# Resilient Backpropagation (Rprop) for Batch-learning in TensorFlow

**Ciprian Florescu & Christian Igel**
Department of Computer Science
University of Copenhagen
2100 Copenhagen Ø, Denmark
ciprianliis@gmail.com, igel@diku.dk

## Abstract

The resilient backpropagation (Rprop) algorithms are fast and accurate batch learning methods for neural networks. We describe their implementation in the popular machine learning framework TensorFlow. We present the first empirical evaluation of Rprop for training recurrent neural networks with gated recurrent units. In our experiments, Rprop with default hyperparameters outperformed vanilla steepest descent as well as the optimization algorithms RMSprop and Adam even if their hyperparameters were tuned.

## 1 Introduction

Wouldn't it be great if neural network training would not require adjusting the hyperparameters of the learning algorithm? For batch learning, this problem is almost fully eliminated by the *resilient backpropagation* (Rprop) algorithms, which are iterative gradient-based optimization methods with adaptive individual step sizes (Riedmiller & Braun, 1993; Riedmiller, 1994; Igel & Hüsken, 2000; Igel & Hüsken, 2003). They are very fast and accurate and their memory requirements scale only linearly with the number of parameters to be optimized. In this paper, we will describe the implementation of Rprop in TensorFlow (Abadi et al., 2015), the most popular software framework for deep learning. The implementation is not straightforward, because some Rprop variants, including our method of choice, require not only information about the current and past gradient, but also about the error on the current and past training batch. Thus, an extension of the standard optimizer interface in TensorFlow is required.

The catch is that the Rprop algorithms excel for batch learning, while many current deep learning applications rely on mini-batch learning. If your data set is large, mini-batch learning offers a much more healthy balance between the time needed for computing an update step and its accuracy. Furthermore, it allows to tailor the learning to the memory constraints of the system, for example, of GPUs. While an Rprop variant for mini-batch learning has already been proposed by Braun (1997) and good results for

---

**Algorithm 1:** iRprop$^+$ algorithm

initialize $\boldsymbol{x}^{(0)} \in \mathbb{R}^n$;
$\forall i \in \{1, \dots, n\} : \Delta_i^{(0)} = \Delta_0 > 0, g_i^{(0)} = 0,$
$\eta^+ > 1, \eta^- \in [0,1]$
$t \leftarrow 1$
**while** *stopping criterion not met* **do**
    $g_i^{(t)} \leftarrow \partial f(\boldsymbol{x}^{(t)})/\partial x_i^{(t)}$ **foreach** $x_i$ **do**
        **if** $g_i^{(t-1)} \cdot g_i^{(t)} > 0$ **then**
            $\Delta_i^{(t)} \leftarrow \min\left(\Delta_i^{(t-1)} \cdot \eta^+, \Delta_{\max}\right)$
            $x_i^{(t+1)} \leftarrow x_i^{(t)} - \text{sign}\left(g_i^{(t)}\right) \cdot \Delta_i^{(t)}$
        **else if** $g_i^{(t-1)} \cdot g_i^{(t)} < 0$ **then**
            $\Delta_i^{(t)} \leftarrow \max\left(\Delta_i^{(t-1)} \cdot \eta^-, \Delta_{\min}\right)$
            **if** $f(\boldsymbol{x}^{(t)}) > f(\boldsymbol{x}^{(t-1)})$ **then**
                $x_i^{(t+1)} \leftarrow x_i^{(t-1)}$
            $g_i^{(t)} \leftarrow 0$
        **else**
            $x_i^{(t+1)} \leftarrow x_i^{(t)} - \text{sign}\left(g_i^{(t)}\right) \cdot \Delta_i^{(t)}$
    $t \leftarrow t + 1$

You can safely set the hyperparameters to their default values, $\eta^- = 0.5$, $\eta^+ = 1.2$, $\Delta_{\min} = 0$, $\Delta_{\max} = \text{large} = 50$, $\Delta_0 = 0.0125$, as done in all all experiments in this study.

---

variant for mini-batch learning has already been proposed by Braun (1997) and good results for

mini-batch learning have been reported elsewhere (e.g., Mosca & Magoulas, 2015; Schuster, 1999), in our experience Rprop – even with slight adaptations – does not perform particularly well in mini-batch scenarios. Still, there are many use cases for batch learning in TensorFlow. Often the number of data points is indeed small or a subset of the available training data is considered for prototyping. We have often been facing this situation when working with recurrent neural networks (RNNs). Furthermore, RNN training is known to be more brittle than training feed-forward architectures. Therefore, we evaluated our Rprop TensorFlow implementation on RNNs for classification and regression. We compared Rprop with default parameters with steepest descent as well as Adam (Kinga & Ba, 2015) and RMSProp (Tieleman & Hinton, 2012), learning methods typically used in TensorFlow.

## 2    RESILIENT BACKPROPAGATION

The Rprop algorithms consider only the signs of the partial derivatives of the function $f$ to be optimized and not their absolute values. In each iteration $t$, every component $x_i^{(t)}$ (e.g., weight) is increased or decreased if the sign of the partial derivative $g_i^{(t)} = \partial f(\boldsymbol{x}^{(t)})/\partial x_i^{(t)}$ is positive or negative, respectively. The amount of the update is equal to the individual step size $\Delta_i^{(t)}$, that is, we have $x_i^{(t+1)} = x_i^{(t)} - \mathrm{sign}(g_i^{(t)}) \cdot \Delta_i^{(t)}$. Before this update, the step size is adapted based on changes of sign of the partial derivative in consecutive iterations. If the partial derivative changes its sign, indicating that a local minimum has been overstepped, then the step size is multiplicatively decreased; otherwise, it is increased. More formally, if $g_i^{(t-1)} \cdot g_i^{(t)}$ is positive then $\Delta_i^{(t)} = \eta^+ \Delta_i^{(t-1)}$, if the expression is negative then $\Delta_i^{(t)} = \eta^- \Delta_i^{(t-1)}$, where $\eta^+ > 1$ and $\eta^- \in [0, 1]$. This procedure is robust w.r.t. the choice of $\eta^+$ and $\eta^-$, which can be fixed to their default values $\eta^+ = 1.2$ and $\eta^- = 0.5$, and therefore the Rprop algorithms require no parameter tuning (e.g., of a learning rate). For RNNs, $\Delta_0$ is set to $0.0125$ (Igel & Hüsken, 2003).

Some Rprop variants implement weight-backtracking. They partially retract "unfavorable" previous steps based on heuristics. In iRprop[+] (Igel & Hüsken, 2003), which is described in pseudo-code in Algorithm 1, a component $x_i^{(t)}$ is reset to its previous value $x_i^{(t-1)}$ iff the signs of $g_i^{(t)}$ and $g_i^{(t-1)}$ differ and the value of the objective function changed to the worse.[1]

For a comparison of iRprop[+] with other Rprop variants, a conjugate gradient method, and the Broyden-Fletcher-Goldfarb-Shanno (BFGS) algorithm, the reader is referred to the article by Igel & Hüsken (2003), which demonstrates the superior performance of iRprop[+] in the domain of neural network training and provides a more detailed description of the algorithm.

## 3    IMPLEMENTATION AND EXPERIMENTAL EVALUATION

**Implementation.**    We implemented the four Rprop variants (Rprop[+], Rprop[−], iRprop[+], iRprop[−]) discussed by Igel & Hüsken (2003) in TensorFlow using C++ and Python.[2] In the following, we focus on the simple and elegant Rprop[−] (Riedmiller, 1994) and iRprop[+]. The latter implements partial weight-backtracking and is in our evaluation the most robust and fastest variant.

**Test problems.**    We consider training RNNs with *gated recurrent units* (GRUs, Cho et al., 2014). The first benchmark was the *Human Activity Recognition using Smartphones* dataset (Anguita et al., 2013). The task is to classify activities based on times signals from motion sensors. The data set contains 10299 instances with a sequence length of 128 and 561 attributes, partitioned into 70% for training and 30% for testing. We used a 2-layer RNN with 48 GRUs per layer. The cross-entropy loss was used for training.

---

[1] After decreasing a step-size parameter, it is ensured that in the next iteration the corresponding step-size is not changed and that backtracking is not applied to the same component two times in a row (this is achieved by setting the variable keeping track of the partial derivative of the $i$th component to zero if the previous update of that variable has been undone, see Algorithm 1).

[2] We will contribute our implementation to the TensorFlow open source repository in the near future.

The second benchmark was predicting the dynamics of the $x$ variable of the Lorenz (1963) attractor five time steps ahead. The RNN had 1-layer with 42 GRUs. We used 500 samples for training and 500 for testing. The squared-error loss was used for training.

**Experimental setup.** We compared with steepest descent, Adam, and RMSPprop. We only considered Rprop for batch learning and with default parameters. In contrast, we optimized the hyperparameters of the other methods using grid-search averaging over 5 trials for each parameter configuration. Learning rates were chosen on a logarithmic scale between $[10^{-6}, 1]$. For Adam, the best two values for $\alpha$ were chosen. For these, all combinations of $\beta_1 \in \{0.5, 0.8, 0.85, 0.9, 0.99, 0.999\}$, $\beta_2 \in \{0.9, 0.99, 0.999, 0.9999\}$ and $\epsilon \in \{10^{-8}, 10^{-6}, 10^{-3}, 10^{-1}, 1\}$ were tested.

As additional baselines, we applied Adam and RMSProp using mini-batches of size 128 for the classification problem, and mini-batches of size 64 for the regression task. For both methods we performed hyperparameter optimization.

**Results.** In our movement classification experiments, batch learning was about 15 times faster than mini-batch learning (NVIDIA Tesla K80, 4 vCPUs, 7.5 GB memory, TensorFlow 1.5.0), due to a bottleneck when computing the aggregated gradients for a multi-layer RNN leading to a worse GPU utilization.

Steepest descent and RMSProp performed worse than Adam for both batch and mini-batch learning. Therefore, they are not discussed any further in this abstract. The best performing parameter configurations for Adam in the movement classification task were ($\alpha = 0.01, \beta_1 = 0.9, \beta_2 = 0.9, \epsilon = 10^{-8}$) and ($\alpha = 0.001, \beta_1 = 0.85, \beta_2 = 0.99, \epsilon = 10^{-6}$) for batch and mini-batch learning, respectively. For Lorenz time series modeling the best parameter configuration for batch learning was ($\alpha = 0.1, \beta_1 = 0.9, \beta_2 = 0.9999, \epsilon = 10^{-3}$) and for mini-batch learning ($\alpha = 0.01, \beta_1 = 0.8, \beta_2 = 0.9999, \epsilon = 10^{-8}$).

Figure 1 shows that on both tasks iRprop$^+$with default parameters outperformed Adam and the other methods even if their hyperparameters were tuned. Furthermore, iRprop$^+$performed better than Rprop$^-$. On the larger movement data set, using mini-batches accelerated Adam with tuned parameters. The higher frequency of updated steps led to faster learning in the beginning, but after 50 epochs iRprop$^+$ gave better results. For both benchmark problems, there were no significant differences on the test data after 300 epochs.

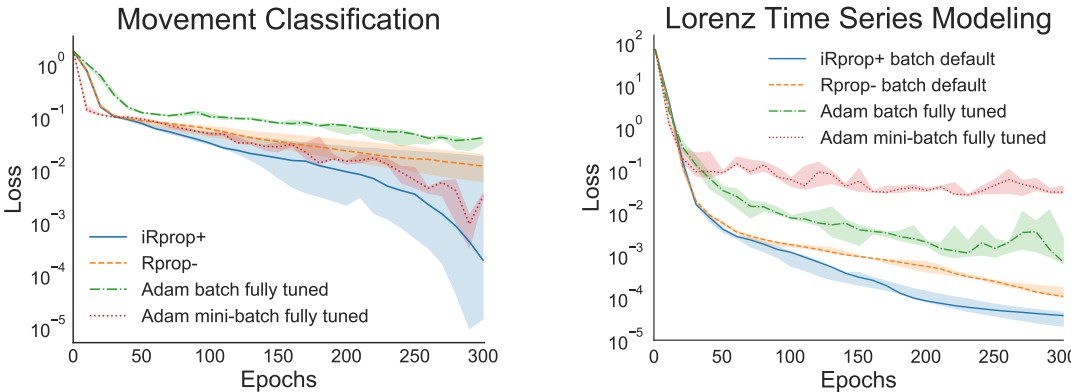

Figure 1: Median of training errors with lower and upper quartiles. Rprop used default parameters, "fully tuned" refers to optimization of all hyperparameters of the learning algorithm.

## 4 CONCLUSIONS

For batch learning in TensorFlow, iRprop$^+$ has become our method of choice and the times of fiddling around with learning rates etc. are over. However, for mini-batch learning (i.e., for large data sets) we still resort to other algorithms.

### ACKNOWLEDGEMENTS

CI acknowledges support from the Innovation Fund Denmark through the *Danish Center for Big Data Analytics Driven Innovation* (DABAI).

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
