# OpenReview forum: "Resilient Backpropagation (Rprop) for Batch-learning in TensorFlow"
_ICLR.cc/2018/Workshop — Accept_

### Official Review · AnonReviewer3 · 2018-03-10

**Rating:** 3
**Confidence:** 5

**Review:**

This paper implemented different variants of Rprop in Tensorflow.

For the implementation, I don't understand why Rprop is harder to implement than other optimizer.

For the experiments, it conclude iRrop+ is the choice for batch learning but the evaluation task isn't standard.

nit, I don't think mini-batch learning is a standard item to refer stochastic gradient decent.

---

### Official Review · AnonReviewer1 · 2018-03-10
**Good paper on resilient backpropagation, more experiments would be welcome.**

**Rating:** 7
**Confidence:** 4

**Review:**

The authors implement the resilient backpropagation (Rprop) algorithm in Tensorflow and show that it performs better than Adam on simple tasks solved with an RNN. This is a good result but more experiments on larger tasks are needed to make the conclusions more stable.

---

### Official Review · AnonReviewer2 · 2018-03-12
**RProp comes back to life**

**Rating:** 7
**Confidence:** 5

**Review:**

This paper describes experiments with Rprop, an older full-batch learning algorithm that depends only on the sign of the gradient in its simplest form, in Tensorflow. The results from the paper show that for the particular problems chosen Rprop works better than RMSprop and Adam. Overall, this is an interesting observation but of course this doesn't mean that every problem out there will benefit from these findings -- more experimental results would be very useful for the reader. I feel more experiments (as described below) would make this study a much more interesting paper for others.

Comments:
- Rprop will work well on full-batch learning but it is not entirely clear that you did that for your experiments -- please clarify what exactly the batchsize is for all your experiments
- Whether full-batch learning is more efficient than mini-batch methods depends heavily on how much data there is total, and also how big your model is. It would be good to compare across a wide set of minibatch sizes (up to full batch) and also for various model sizes given very large amounts of training data to make useful recommendations for others. For very large amounts of data full-batch methods are likely always losing against methods using smaller batches because of the greatly reduced number of updates (you mention something along these lines as well).
- The PhD thesis "On supervised learning from sequential data with applications for speech recognition" describes another version of Rprop (ARprop) that seems to have worked quite well for RNNs even for larger amounts of data -- probably worth trying or at least mentioning as well.
- Please use easier to identify line styles in your graphs, it is not clear which line belongs to which method
- Your TL;DR for the description of the paper sounds a bit too confident:  "TL;DR: For batch learning in TensorFlow, Rprop is the method of choice and the times of fiddling around with learning rates etc. are over.", maybe adjust this message a little bit since it depends on how much data you have etc.

---

### Decision · Program_Chairs · 2018-03-20
**ICLR 2018 Workshop Acceptance Decision**

**Decision:**

Accept

**Comment:**

Congratulations, your paper was accepted to the ICLR workshop.